# npphen: An R-Package for Detecting and Mapping Extreme Vegetation Anomalies Based on Remotely Sensed Phenological Variability

Roberto O. Chávez [1,2,3,*], Sergio A. Estay [4,5], José A. Lastra [1], Carlos G. Riquelme [5], Matías Olea [1], Javiera Aguayo [1] and Mathieu Decuyper [6]

1   Laboratorio de Geo-Información y Percepción Remota, Instituto de Geografía, Pontificia Universidad Católica de Valparaíso, Valparaíso 2362807, Chile
2   Millenium Nucleus in Andean Peatlands (AndesPeat), Avenida 18 de Septiembre 2222, Arica 1000965, Chile
3   Institute of Ecology and Biodiversity, Las Palmeras 3425, Ñuñoa, Santiago 7800003, Chile
4   Instituto de Ciencias Ambientales y Evolutivas, Facultad de Ciencias, Universidad Austral de Chile, Valdivia 5090000, Chile
5   Center of Applied Ecology and Sustainability, Facultad de Ciencias Biológicas, Pontificia Universidad Católica de Chile, Santiago 7510177, Chile
6   Forest Ecology and Management Group, Wageningen University, Droevendaalsesteeg 3, 6708 PB Wageningen, The Netherlands
*   Correspondence: roberto.chavez@pucv.cl

**Abstract:** Monitoring vegetation disturbances using long remote sensing time series is crucial to support environmental management, biodiversity conservation, and adaptation strategies to climate change from global to local scales. However, it is difficult to assess whether a remotely detected vegetation disturbance is critical or not, since available operational remote sensing methods deliver only maps of the vegetation anomalies but not maps of how "uncommon" or "extreme" the detected anomalies are based on the available records of the reference period. In this technical note, we present a new release of the probabilistic method and its implementation, the npphen R package, designed to detect not only vegetation anomalies from remotely sensed vegetation indices, but also to quantify the position of the anomalous observations within the historical frequency distribution of the phenological annual records. This version of the R package includes two new key functions to detect and map extreme vegetation anomalies: *ExtremeAnom* and *ExtremeAnoMap*. The npphen package allows remote sensing users to detect vegetation changes for a wide range of ecosystems, taking advantage of the flexibility of kernel density estimations to account for any shape of annual phenology and its variability through time. It provides a uniform statistical framework to study all types of vegetation dynamics, contributing to global monitoring efforts such as the GEO-BON Essential Biodiversity Variables.

**Keywords:** disturbance; remote sensing; time series; climate change; EBV; GEO-BON



## 1. Introduction

Reliable and timely information regarding the status of vegetation ecosystems is key to achieve the ambitious conservation goals acknowledged by the parties of the Convention on Biological Diversity [1,2]. Satellite remote sensing has contributed enormously to the understanding of vegetation dynamics affected by climate change [3–6]. Systematically acquired satellite records allow scientists to assess the effects of local to large-scale disturbances such as fires [7,8], volcano eruptions [9], insect outbreaks [10–12], or climate extremes [13–15]. The general approach for the multitemporal remote sensing assessments of vegetation disturbances is to set a phenological baseline and account for anomalies or deviation from this baseline [16]. With the opening of long time series satellite archives, such as Landsat, NOAA-AVHHR, MODIS, VIIRS, GOES- 16 and 17, and the ESA-Sentinels,

the development of vegetation anomaly detection methods based on the time series of satellite images has rapidly developed, allowing for more comprehensive vegetation disturbance monitoring and recovery [17,18]. These time series analysis methods rely on a robust estimation of the annual phenological cycle as a dynamic baseline and therefore vegetation phenology and anomaly detection are closely related remote sensing research subjects [16,19,20].

Vegetation indices based on optical remote sensing data such as the Normalized Difference Vegetation Index or NDVI [21] and the Enhanced Vegetation Index or EVI [22] time series provide good approximations of the annual phenological development of vegetation cover or "greenness" at different temporal and spatial resolutions [23] and are included in standard satellite monitoring products such as the MODIS MOD13Q1 and MYD13Q1 from the Terra and Aqua satellites [24] with more than two decades of consistent and continuous records. Several methods for vegetation anomaly detection have been developed to exploit this wealth of remote sensing data: the "phenology" [25], "phenopix" [26] and "greenbrown" [27] R packages provide functions to interpolate or fit parametric phenological curves to data for a single growing season or consecutive growing seasons from which departures or anomalies can be calculated for a given monitoring period. The "bfast" R package decomposes the series into a phenological (seasonal) component, a trend component, and a remainder error [28]. In this method, abrupt changes in trends can be related to vegetation perturbations, defining periods (between perturbations) for which the phenological signal is homogeneous and accounted for by the parametric function. For all parametric methods, the theoretical framework works by assuming that regular, annual curves are a good representation of the plant phenological cycle (e.g., [29,30]), which is not valid for all types of vegetation, for instance in arid or semi-arid ecosystems where green-ups of vegetation follow unpredictable non-seasonal rainfall [31–33]. Moreover, significantly different results can be obtained on phenometric estimations when using different interpolation techniques [23]. However, parametric approaches are the most used in the literature, see [23,34] for reviews of methods for land surface phenology using remote sensing.

In our package, we propose a different approach that uses the observed frequency values of the vegetation index, and defines the expected distribution directly from observed data without reference to a theoretical model. The advantage of this approach is its flexibility to adapt to the particular phenological cycles of different vegetation, e.g., tropical forests or arid and semi-arid ecosystems where seasonal approaches are not suitable. The non-parametric approach was first implemented in the "npphen" R package (version 1.1-0) in 2017 to quantitatively describe the annual phenological variability of vegetation, from which anomalies can be calculated. Although "npphen" and some derivatives of this algorithm have already been used in different applications and ecosystems (e.g., [12,15,35,36]), the key R functions to detect and map extreme vegetation anomalies have not been published in the scientific literature. In this technical note, we present updated and new functionalities of the "npphen" package which has been recently released (13 October 2022) on the CRAN in the R environment ("npphen" 1.5.2). The new functions that have been incorporated are the capability of the algorithm to quantify how unlikely or uncommon the anomalies are, based on the historical frequency of remotely sensed vegetation indices, which are now available for both numerical vectors (*ExtremeAnom*) and raster stacks (*ExtremeAnoMap*). Nevertheless, we show in this contribution all the functions of the latest release of "npphen". We first introduce the mathematical basis of the method, followed by two examples showcasing the extreme anomaly mapping functions with positive and negative (drought) extreme vegetation disturbances. Both events have not been reported before and are original study cases. Finally, we provide examples using different remote sensing data (MODIS, Landsat and Sentinel2).

## 2. Description of the Method

### 2.1. Organization of the R Package

The current release of the "npphen" R package includes five functions, which can be organized into two branches (Figure 1): numerical vector functions and raster stack functions. Similar phenology functions and anomaly functions are available for both numerical vectors and raster stacks. While the phenology functions are devoted to describing and mapping the variability of the annual phenological cycle for the baseline period (see Section 2.1), the anomaly functions are devoted to accounting for anomalies and assessing them on "how uncommon" they are in a continuous way (see Section 2.2). Both the phenology and anomaly functions are designed to work with time series of vegetation indices as inputs in the format of numerical vectors or raster stacks. Examples of input numerical vectors can be an NDVI time series of a single pixel of satellite data or a time series of the green index (R/[R + G + B]) calculated on landscape pictures from Phenocams [37]. Examples of input raster stacks can be a MODIS Terra NDVI time series from MOD13Q1 16-day composites or daily GOES NDVI time series of specific locations, countries, continents or for the entire world. In order to deal with high level computing tasks, the functions using raster stacks as input data are designed to work in parallel with multi-core capabilities, taking full advantage of multi-core servers or virtual machines such as the EC2 of Amazon Web Services. In order to optimize heavy computing loads, the different functions are designed to use integer values as input data. For this reason, if an NDVI time series has values ranging from −1.0 to 1.0, it is advised that users multiply by 10,000 to acquire a −10,000 to 10,000 range before applying the "npphen" functions.

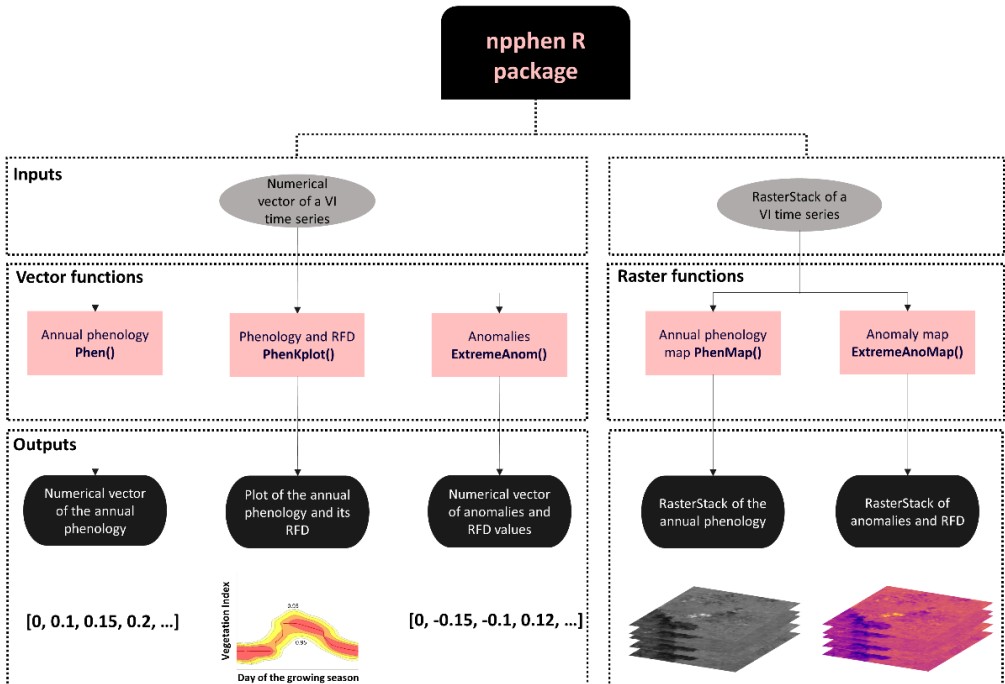

**Figure 1.** Inputs, functions and outputs for the "npphen" R package. Both phenology functions and anomaly functions can use either numerical vectors or raster stacks as inputs. Consequently, the outputs are numerical vectors or raster stacks, except for the PhenKplot function whose output is a graph of the reference phenology along with its historical variability.

### 2.2. Calculation of the Annual Phenological Baseline and Its Variability

To detect vegetation anomalies using a time series of a vegetation index, such as NDVI (e.g., Figure 2a–f), the first step is to define the annual phenological baseline. In our approach, we estimate the most expected value of a vegetation index through the growing season based on the probability density function $f(x)$. We approximate $f(x)$ by $\hat{f}(x)$ using a

Kernel Density Estimation (KDE) procedure. This is calculated from a time series, defined as X = (X1, . . . , Xn), containing paired values of a vegetation index (VI) and time (dates). See Figure 2a–f for some examples of VI time series. The dates are then converted to days of the growing season (DGS), ranging from 1 to 365 (Equation (1)).

$$X_i = (DGS_i, VI_i)^T, \text{ where } i = 1, \ldots, n \tag{1}$$

For example, if our dataset covers p annual phenological cycles with m observations per cycle, then our time series will contain n=m × p observations. The npphen algorithm does not require the same number of observations per cycle.

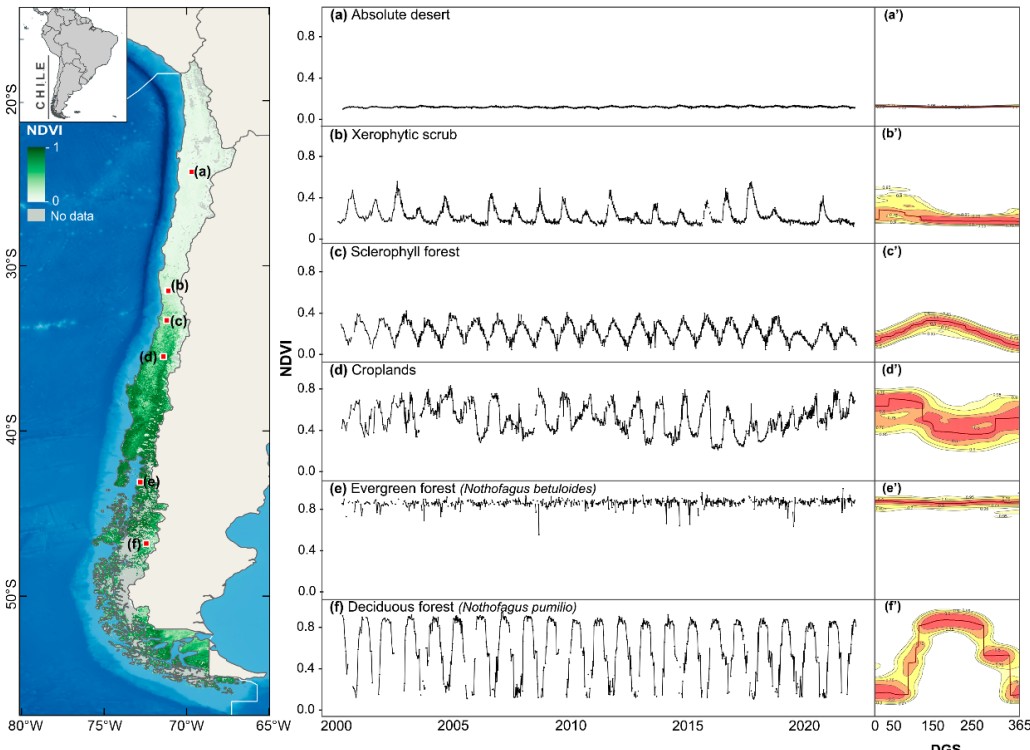

**Figure 2.** NDVI time series at different locations in Chile (**a**–**f**) and kernel density estimations (KDE) of the annual phenology (**a′**–**f′**) considering 22 years of MOD/MYD13Q1 NDVI records from the Terra and Aqua satellites. The KDE shows in yellow to red gradient colors the frequency of NDVI values at different days of the growing season (DGS). NDVI values in the map on the left correspond to a MYD13Q1 NDVI image of January 2022 (the peak of the growing season).

We define $\hat{f}$(x) as the bivariate density function of X estimated by Equation (2) as follows:

$$\hat{f}(x; H) = \frac{1}{n} \sum_{i=1}^{n} K_H(x - X_i) \tag{2}$$

where x is a generic point in the bivariate DGS-VI space, Xi = (DGS$_i$, VI$_i$)$^T$, H is the bandwidth 2 × 2 matrix, and K is the kernel. The bandwidth matrix H defines the size of the kernel in each dimension (diagonal) and the rotation of the kernel in reference to the axes (anti-diagonal). KDE in 2 dimensions works by centering a bivariate kernel (e.g., a Gaussian kernel) around each observation, and by averaging the heights of all kernels till obtaining the final density estimation. The size of the kernel in each dimension is defined by H. For more details about theoretical aspects of KDE see [38].

In our algorithm, we used a Gaussian kernel over the observed VI values. The estimation of the VI frequencies is performed over a grid of 365 columns (daily estimation) and 500 rows. The bandwidth H is not fixed and is defined using the multivariate plug-in

selector of Wand and Jones (1994). Using $\hat{f}(x)$ we can identify the most frequent values of VI along the phenological cycle (the reference annual phenology for this site) and its frequency distribution (phenological variability along the years). Figure 2a′–f′ shows the flexibility of the method in describing the reference phenological curve and its variability for a wide range of ecosystems ranging from the Atacama Desert (Northern Chile) to sub-antarctic forests (Southern Chile). The "PhenKplot" function was used to obtain the plots of Figure 2a′–f′.

### 2.3. Anomaly Detection and Assessment Based on the Phenological Reference Frequency Distribution (RFD)

Anomalies are calculated as the difference between the observed VI value ($VI_{obs}$) and the most frequent or expected VI value ($E(VI_i)$), according to the historical frequency distribution (Equation (3)).

$$A_i = VI_{obs} - E(VI_i) \qquad (3)$$

Besides the magnitude of the either positive or negative anomaly, in our method, the position of the VI observation within the reference frequency distribution (RFD) is recorded. This value, ranging from 0.0 to 0.99, indicates how extreme the anomaly ($A_i$) is. An $A_i$ between 0.90 and 0.99 RFD can be considered to be an extreme anomaly, and can be defined by the user. Figure 3 provides an example of extreme VI anomalies of *Nothofagus macrocarpa* forests caused by a severe drought which occurred in Central Chile during 2019 and detected using the "npphen" R package.

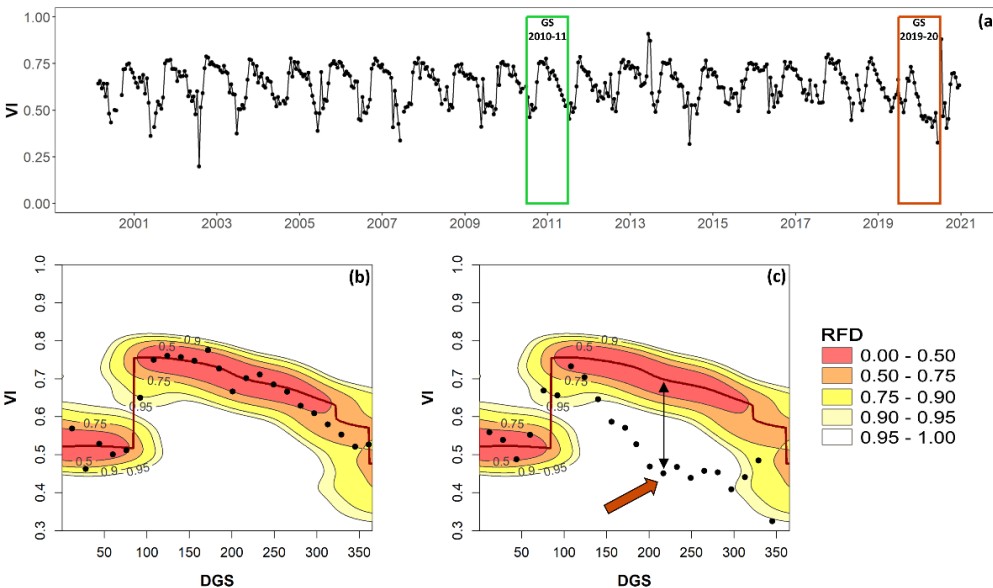

**Figure 3.** Detection of anomalies (Ai) outside the 95% of the reference frequency distribution (RFD >= 0.95) using the "npphen" R package. (**a**) A vegetation index (VI) time series of a *Nothofagus macrocarpa* forest in Central Chile where a severe drought occurred during the growing season (GS) 2019–20 (red box). (**b**) The reference frequency distribution (RFD) of all VI observations is calculated by means of kernel density estimations in the VI–DGS space. Black dots are VI observations of the GS 2010–11 with all records falling inside RFD < 0.95, i.e., a normal year. (**c**) VI observations during the GS 2019–20 showing values at RFD >= 0.95, i.e., with extreme negative VI anomalies as a consequence of the severe drought.

Figure 3a shows a VI time series of a single pixel of deciduous *Nothofagus macrocarpa* forest in Central Chile. The green box indicates a normal growing season (2010–11) and the red box the growing season of the 2019–20 drought. In this example, the entire time series was used to calculate the RFD. In Figure 3b, we observed that all VI records of the 2010–11 growing season (black dots) fall inside the 0.95 RFD, and therefore no extreme anomalies

are flagged. On the contrary, Figure 3c shows how the VI was increasing during spring to reach its maximum value around DGS 110 to gradually drop as the drought conditions were affecting the forest canopy. By DGS 220 the VI records resembled the VI values of Winter. In this case, a period of extreme negative vegetation anomalies can be observed between DGS 150 and 300 where the VI records were outside the 0.95 RFD.

This way, our method provides two key outputs for anomaly detection: the magnitude of the anomaly ($A_i$) and the location within the RFD ($RFD_i$). These are the outputs of the *ExtremeAnom* function for numerical vectors and the *ExtremeAnoMap* function for raster stacks (Figure 1). These are calculated per DGS and per pixel in the case of raster stacks. For the *ExtremeAnom* and *ExtremeAnoMap* functions, users can set the "output" argument as "both" (the default option) to obtain $A_i$ and $RFD_i$ together as outputs, in which case the first "i" outputs will correspond to the Ai anomalies and the next "i" will correspond to the $RDF_i$ positions. The "output" argument can also be set as "rfd" to obtain only the $RDF_i$ positions. The third and last option is to set "output" as "clean" to obtain only the Ai anomalies whose RFD position overpass a given threshold that the user defines as the "extreme condition" (e.g., 0.95). This critical threshold is set by the user using the "rfd" argument, (e.g., RFD = 0.95). It is important to note that a threshold of 0.95 RFD implies that the anomaly detection will consider extreme VI values in the two tails of the RFD, i.e., among the 2.5% of the highest and 2.5% of the lowest recorded VI observations of the reference period.

A detailed tutorial on the use of the "npphen" functions applied to the case of the insect outbreak is available at: https://www.pucv.cl/uuaa/labgrs/proyectos/introduction-to-npphen-in-r, accessed on 3 October 2022.

## 3. Examples of Extreme Vegetation Anomaly Detection and Mapping

### 3.1. Extreme 2019 Drought in Central Chile

In this first example (Figure 4), we used the ExtremeAnoMap function and MOD13Q1 NDVI data to detect extreme negative NDVI anomalies caused by a severe drought in Central Chile [39]. Central Chile has faced a so-called Mega Drought since 2010, and therefore, the reference period for anomaly calculations was set to 2000–2009. All anomalies and RFD positions for the period of 2010–2022 were calculated, but we show only the results of the date when the effects of the drought on vegetation were most extensive (spring 2019, when about 20,431 km$^2$ of vegetation showed RDF >= 0.95). Figure 4a shows the NDVI anomalies calculated by the end of September 2019, with drought affecting the vegetation intensively in the south of the Coquimbo region, most of the Valparaíso region and the Northern part of the Santiago metropolitan region. This part of the country has the highest density of the Chilean population with more than 10 million inhabitants. Figure 4b shows the RFD position of the NDVI observations at that time, with a clear cluster of RFD >= 0.95 at these three administrative regions, indicating where the NDVI values at that time fall within the 2.5% of the lowest records of the reference period. In this way, the extreme (RFD >= 0.95) negative NDVI anomalies could be identified and mapped (Figure 4c).

No validation datasets are available to contrast the detected extreme vegetation anomalies for Central Chile using the "npphen" R package. However, part of the authors' team is promoting the development of a Phenocam network and, as part of the testing phase of the system, we captured some photographs of the condition of a *Nothofagus macrocarpa* forest (Figure 5), located in the La Campana National Park in Central Chile (71.12°S, 32.96°W), during the 2019–20 drought. In this Figure, we can observe that for the peak of the 2019–20 growing season (summer), the canopy condition (Figure 5b) was similar to the one of autumn (Figure 5c), which is in line with the extreme anomaly detection using MODIS data and the "npphen" R package at that time of the year (Figure 5a).

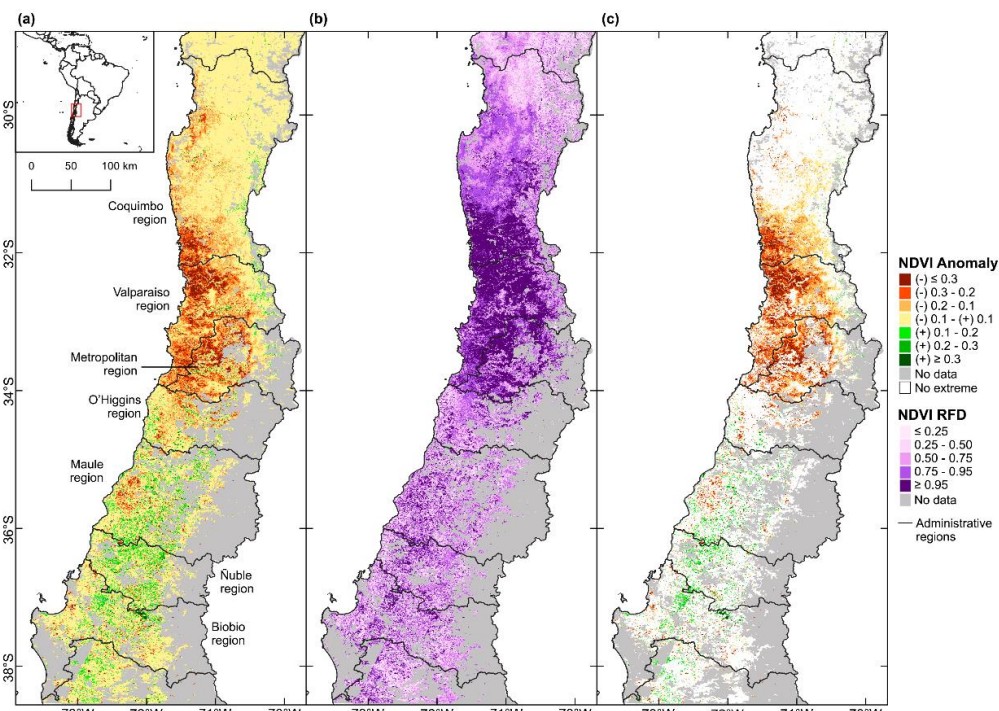

**Figure 4.** Extreme negative anomalies of the Normalized Difference Vegetation Index (NDVI) caused by an extreme drought in Central Chile. Calculations were performed using the *ExtremeAnoMap* function of the "npphen" R-package and MOD13Q1 data from the Terra satellite and considered 239 NDVI scenes of the 2000–2009 period as a reference. The maps correspond to 30 September 2019 and show (**a**) NDVI anomalies, (**b**) the position of the NDVI observations within the reference frequency distribution (RFD), and (**c**) NDVI anomalies at RFD >= 0.95. The "extreme" NDVI anomalies shown in panel (**c**), defined by RFD >= 0.95, are NDVI records falling among the 2.5% range of the lowest NDVI values of the reference period.

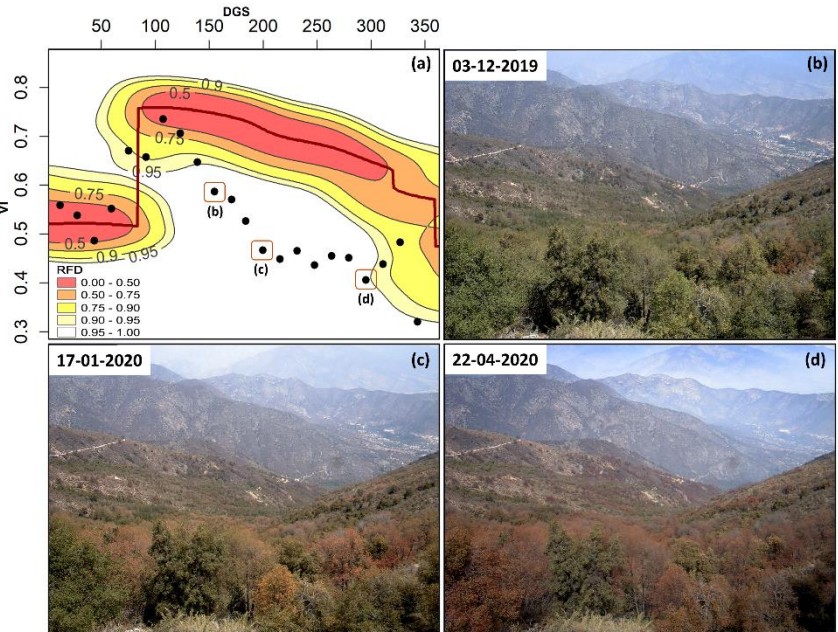

**Figure 5.** Phenocam photographs of *Nothofagus macrocarpa* forest in Central Chile (**b**–**d**) during the extreme 2019–20 drought (**a**). Note that the condition of the forest canopy by January 2020 (summer) resembles the typical reddish colour of Autumn.

### 3.2. Extreme Greening in Central and Northern Chile of 2017

After six years of drought in a row, 2017 experienced abnormally high precipitations in Central and Northern Chile related to the so-called "coastal El Niño" [40], that even caused a "blooming" desert event in Atacama with a massive greening up and flowering of desert species. In this example, we used the same reference period as the previous example, and the NDVI anomalies (Figure 6a) filtered by RFD >= 0.95 (Figure 6b) provide the location and intensity of the "greening event" of 2017 (Figure 6c), defined as extreme NDVI positive anomalies, i.e., observed NDVI values falling within the 2.5% of the highest NDVI values recorded during the reference period. At this particular date (26 June 2017), corresponding to the peak of the "greening", an area of 36,660 km² showed an extreme positive anomaly. It is also remarkable to note the extreme negative anomalies (circled area) observed in the El Maule region as a consequence of the mega-fire or "tormenta de fuego", when hundreds of fires took place simultaneously, burning more than 5000 km² of natural vegetation and plantations of the Maule region [8].

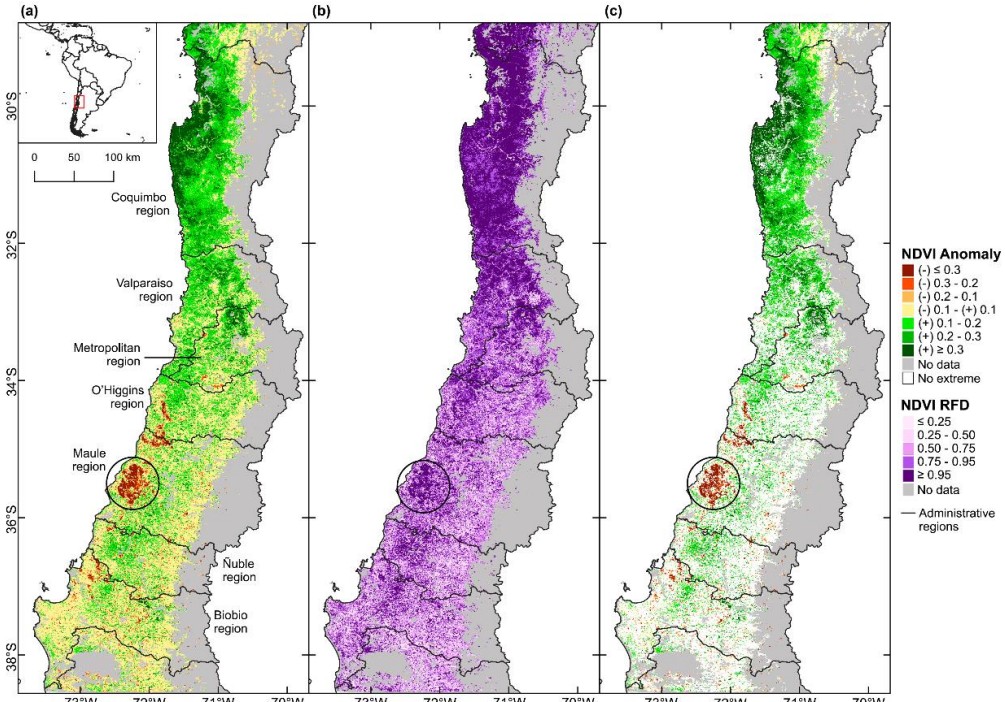

**Figure 6.** Extreme positive anomalies of the Normalized Difference Vegetation Index (NDVI) related to abnormally high precipitation in Central and Northern Chile. Calculations were performed using the *ExtremeAnoMap* function of the "npphen" R-package and MOD13Q1 data from the Terra satellite and considered 239 NDVI scenes of the 2000–2009 period as a reference. The maps correspond to 26 June 2017 and show (**a**) NDVI anomalies, (**b**) the position of the NDVI observations within the reference frequency distribution (RFD), and (**c**) NDVI anomalies at RFD >= 0.95. The "extreme" NDVI anomalies shown in panel (**c**), defined by RFD >= 0.95, are NDVI records falling among the 2.5% range of the highest NDVI values of the reference period. The circled area corresponds to the extreme negative anomaly due to the mega-fire of January 2017, known as "tormenta de fuego".

### 3.3. Examples Using Different Remote Sensing Data

Finally, we provide examples of extreme anomaly detection for the case of the 2019 drought (Section 3.1) using NDVI data with different spatial and temporal resolutions: MODIS 13 Q1 with 250 m spatial resolution and 16-day composites (i.e., daily cloud-free data aggregated to 16 days), Landsat with 30 m spatial resolution and 16-day temporal resolution (therefore with more temporal gaps), VIIRS with 500 m spatial resolution (as it is available at the LP DAAC geoportal; native resolution is 375 m) and 8-day composites (similar to MODIS, daily cloud-free data aggregated to 8 days), and Sentinel 2ab with 10 m

spatial resolution and 5-day temporal resolution (when combining a and b). For all datasets, we selected a single pixel of native forest, specifically a deciduous forest of *Nothofagus macrocapa*, the same used in Figures 3 and 5, and retrieved the NDVI time series using the Google Earth Engine capabilities.

The left panels in Figure 7 show the NDVI time series depicting the seasonal behavior of the deciduous forest. The right panels show the KDE reconstruction of the annual phenology and its variability. Readers can observe in black dots the NDVI values of the GS 2019–2020, falling outside the 0.95 RFD for all four datasets, which is an indication of extremely negative effects. Therefore, the npphen R-package in combination with different datasets allows the detection of extreme negative NDVI anomalies related to the 2019 drought. This illustrates the capability of the method to work with different datasets, temporal resolutions, temporal gaps, and time frames.

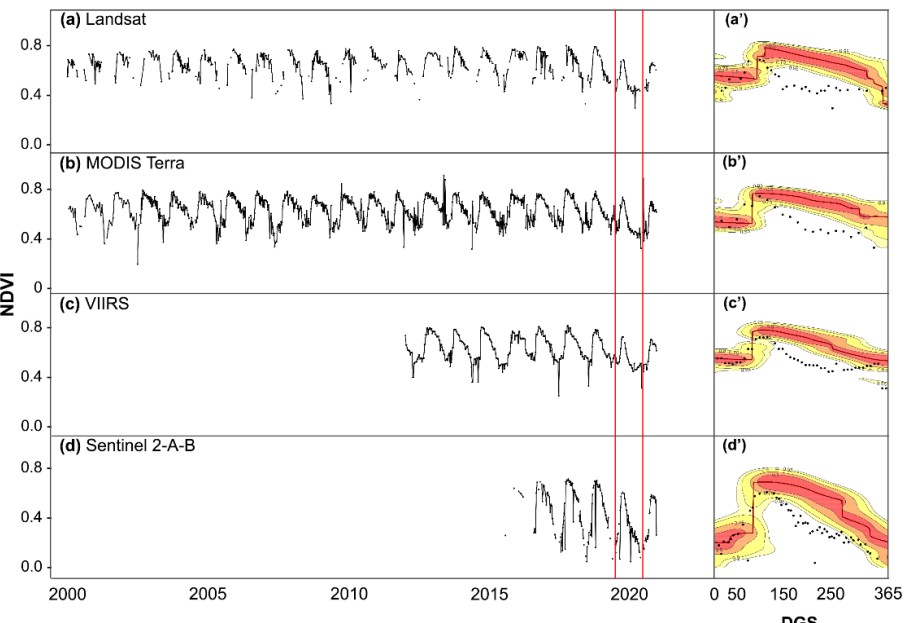

**Figure 7.** NDVI time series of a single pixel of deciduous *Nothofagus macrocarpa* native forest in Central Chile (71°W, 33°S) from four different remote sensing products (**a–d**). Since all datasets have different time frames, for comparison purposes, we set as a reference period all dates except for the GS 2019–2020 (between the red lines) when the extreme drought described in Section 3.1 took place. In all cases, the reference phenology and its variability could be described by the KDE estimations (**a'–d'**) and extreme negative detected in 2019.

## 4. Discussion and Concluding Remarks

Monitoring the status of biodiversity by integrating different data sources in a standardized way is one of the major challenges for intergovernmental efforts such as the Group on Earth Observation- Biodiversity Observation Network or GEO-BON (www.geobon.org, accessed on 10 December 2022), which is promoting a harmonized system to retrieve and report on the status of Essential Biodiversity Variables (EBVs). In line with this ambitious goal, the "npphen" R-package provides a unified operational algorithm capable of creating a phenological baseline for any type of vegetation beyond their particular annual behavior (Figure 2a'–f'). This can be useful to build up the phenological baseline of all vegetation types of a specific area of interest or a network, such as the long-term ecological research sites (e.g., https://lter-europe.net/, accessed on 10 December 2022), using a single statistical approach. Although the package was initially designed to process remotely sensed vegetation indices data, it may also be a good alternative to analyze any kind of temporal records, e.g., the green index from Phenocams, Flux Towers measurements, or climatic records from meteorological stations.

Since no parametric curves are used to model the expected annual phenological cycle, as other algorithms do, the representation of the phenological baseline and the anomaly detection are not hindered by the quality of the curve fitting procedure. This makes "npphen" not only flexible to describe all possible annual phenological cycles of vegetation, but also to account for the inter-annual variability. This feature is useful to assess how "extreme" the detected anomalies are by checking their magnitude, either negative or positive, in the framework of the recorded historical frequency. Outliers (e.g., due to cloud contamination) within the time series are not critical for the reference period, especially for a longer time series (e.g., >10 years), but are indeed important for the detection period since it has a direct effect on the anomaly calculation and its RFD position for that particular date. Therefore, users should apply the QA bands of the different products to neglect unusable data. Another option is to consider consecutive records with RFD >= 0.95 to flag true anomalies, which can be achieved by post-processing the outputs of the *ExtremeAnoMap* function.

**Author Contributions:** R.O.C. and S.A.E. conceived the original code and package development. J.A.L., M.O., C.G.R. and M.D. provided specific R code contributions to the current package version. J.A.L. prepared and submitted the current R package version to Github and CRAN. J.A.L., J.A. and M.O. prepared the example datasets, figures and tutorials. R.O.C. prepared the manuscript. All authors have read and agreed to the published version of the manuscript.

**Funding:** This research was funded by Fondecyt regular N° 1211924, Fondecyt regular N° 1201714, Fondef IDeA I+D 2021 ID21|10249, ANID-MILENIO-NCS2022_009 and Grant ANID PIA/BASAL FB210006 (ROC) and by ANID PIA/BASAL FB0002 and Fondecyt regular N° 1211114 (SAE).

**Data Availability Statement:** The current stable version of the package (1.5.2) can be downloaded directly from the CRAN or from the following URL: https://github.com/labGRS/npphen (accessed on 10 December 2022). It can be installed with this command line: remotes::install_github('labGRS/npphen').

**Conflicts of Interest:** The authors declare no conflict of interest. The funders had no role in the design of the study; in the collection, analyses, or interpretation of data; in the writing of the manuscript; or in the decision to publish the results.

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
