# Peer review of "npphen: An R-Package for Detecting and Mapping Extreme Vegetation Anomalies Based on Remotely Sensed Phenological Variability"

_remotesensing, doi:10.3390/rs15010073_

Round 1

Reviewer 1 Report

The authors presents the npphen R package - a tool for time series analysis of remote sensing data for detection of vegetation disturbances. The npphen package has been already published and used by several studies (12, 15, 35 and 36 in the reference list). Also the two out of three showcases has been already published in Estay and Chavez, 2018 (https://doi.org/10.1101/301143). Authors advertised that new functions are introduced, namely to quantify how unlikely the anomalies are. If I'm not mistaken, very similar (or even the same) concept of anomalies likelihood has been already published in the AVOCADO algorithm (Decuyper et al., 2022, https://doi.org/10.1016/j.rse.2021.112829).

So in principle, i like the compact presentation of the npphen R package, it is clearly written and supported by three show cases, but I'm not convinced about the novelty of such a contribution and if it deserves an independent research paper, when most of its components has been previously published.

Maybe a comparison with other anomaly detection algorithms could be the newly added value here. Would it be feasible to expand the third show case and process the data with the other tools and compare the results with npphen?

Author Response

Dear reviewer

Thank you for your time, comments and suggestions. We prepared a PDF with the response. Please see the attachment.

Best regards. 

Dr Roberto O. Chávez

Reviewer 2 Report

I’m very pleased to inform that article: “npphen: an R-package for detecting and mapping extreme vegetation anomalies based on remotely sensed phenological variability” can be accept in present form. Manuscript shows interesting R-package: npphen for analysing anomalies in vegetation without additional theoretical assumptions. With additional case studies it fit aim and scope of the Remote Sensing journal.

Only ground truth validation (or cross validation based on different approach)  might be added to the case studies shown in the article.

Author Response

(The authors gave the same response as above.)
